ALDOB is a prognostic biomarker and a potential immunotherapy target for clear cell renal cell carcinoma

Xu Wu
Wu Dali
Li Cuilian
Yan Lingfei
Peng Bo
Luo Yang
Liu Dawei
Li Qing liq73@163.com
Wang Tao smu02204633@i.smu.edu.cn
Department of Urology, The Fifth Affiliated Hospital, Southern Medical University , Guanzhou , China
Soares Paula
Electronic publication date: 2025 Aug 18
Publication date: 2025
Volume: 13
Electronic Location ID: e19869
Received 2024 Dec 20; Accepted 2025 Jul 17
Copyright: ©2025 Xu et al.
Copyright year: 2025
Copyright holder: Xu et al.
License: This is an open access article distributed under the terms of the Creative Commons Attribution License, which permits unrestricted use, distribution, reproduction and adaptation in any medium and for any purpose provided that it is properly attributed. For attribution, the original author(s), title, publication source (PeerJ) and either DOI or URL of the article must be cited.
License URL: https://creativecommons.org/licenses/by/4.0/

Keywords: ALDOB, Prognosis, Tumor immune microenvironment, Clear cell renal cell carcinoma, TCGA, GEO, HPA, Immunohistochemistry, UALCAN, TIMER

Funding: Characteristic Innovation Program of Ordinary Colleges and Universities of Guangdong Province of China 2024KTSCX076 Project of Administration of Traditional Chinese Medicine of Guangdong Province of China 20251263 Medical Scientific Research Foundation of Guangdong Province of China B2024032 President Foundation of The Fifth Affiliated Hospital, Southern Medical University YZ2023ZX08 This work was funded by Characteristic Innovation Program of Ordinary Colleges and Universities of Guangdong Province of China (2024KTSCX076); the Project of Administration of Traditional Chinese Medicine of Guangdong Province of China (20251263); Medical Scientific Research Foundation of Guangdong Province of China (B2024032); the President Foundation of The Fifth Affiliated Hospital, Southern Medical University (YZ2023ZX08). The funders had no role in study design, data collection and analysis, decision to publish, or preparation of the manuscript.

==============================
Background

Aldolase B (ALDOB), functioning as a glycolytic enzyme, exhibits a controversial role in malignancies and demonstrates dual potential as both a tumor suppressor and cancer-promoting enzyme. Nevertheless, it is still uncertain if there is a relationship between ALDOB levels, prognosis, and tumor-infiltrating lymphocytes in clear cell renal cell carcinoma (ccRCC).

Objective

This study aims to investigate the prognostic significance of ALDOB in ccRCC and its potential association with clinicopathological features and tumor immune microenvironment. By integrating multi-database bioinformatics analysis and experimental validation, we seek to elucidate the role of ALDOB in ccRCC progression and its potential as a predictive biomarker.

Methods

To ascertain the potential link between ALDOB level, clinical parameters, and overall survival (OS) in individuals with ccRCC, we employed diverse databases, which include The Cancer Genome Atlas (TCGA), Gene Expression Omnibus (GEO), the Human Protein Atlas (HPA) and The University of Alabama at Birmingham Cancer data analysis Portal (UALCAN). Furthermore, an in-depth analysis of the link between tumor-infiltrating immune cells (TIIC) and ALDOB was carried out using the TIMER database. Immunohistochemistry (IHC) was applied to identify the ALDOB level in a tissue microarray.

Results

The expression of ALDOB demonstrated a strong association with pathologic T stage, pathologic N stage, pathologic M stage, histologic grade, and gender. Decreased ALDOB level was linked to unfavorable disease-specific survival (DSS), progress free interval (PFI), and OS outcomes (p < 0.001). Subsequently, a marked link was observed between ALDOB level and a heightened presence of infiltrating Treg, Th17 cells, and neutrophils in ccRCC. IHC showed that the ALDOB level in ccRCC samples was notably diminished relative to that in the adjacent normal tissues.

Conclusions

As a prospective predictive indicator for individuals with ccRCC, reduced ALDOB level exhibited strong correlations with clinical characteristics, unfavorable outcomes, and immune infiltration in individuals with ccRCC.

Introduction

Renal cell carcinoma (RCC) is the most lethal urological malignancy in terms of annual mortality, and its global prevalence has been increasing (Young et al., 2024). Clear cell renal cell carcinomas (ccRCCs) represent the predominant pathological types of RCCs with a poorer prognosis than papillary RCCs and chromophobe RCCs, accounting for 70% to 80% of all cases of RCCs (Lai et al., 2021). The majority of ccRCCs exhibit early inactivation of the tumor suppressor gene von Hippel-Lindau (VHL) (Page et al., 2025). Despite targeted therapy being one of the most commonly used conventional therapies to treat ccRCC, almost all patients eventually experience deterioration of their condition because ccRCC cells are resistant to drug-induced apoptosis (Sánchez-Gastaldo et al., 2017; Serzan & Atkins, 2021). Ferroptosis induction, a novel form of cell death, is emerging as a potential alternative therapeutic approach for ccRCC (Pan et al., 2024; Zou et al., 2019). Currently, existing therapies only show promise for a limited portion of ccRCC patients, making it imperative to find more effective therapeutic targets. An additional pressing priority remains to discover novel biological indicators for early ccRCC identification and enhanced outcome prediction.

ALDOB encodes an enzyme that converts fructose 1-phosphate to dihydroxyacetone phosphate and glyceraldehyde (Tang & Cui, 2024). Malignancies display aberrant expression of ALDOB, which serves a function not only in glycolysis and fructose metabolism, but also in tumorigenesis. These abnormal expression patterns are highly related to cancer patients’ clinicopathological features and prognosis. Researchers found that elevated ALDOB expression exhibits strong links to clinical aspects of rectal adenocarcinomas, encompassing tumor progression and lymphovascular infiltration (Tian et al., 2017). In addition, the survival analysis revealed that patients with rectal cancer who expressed high levels of ALDOB had worse disease-specific survival (DSS). This might serve as a prognostic biomarker. In contrast, investigations into gastric cancer revealed diminished ALDOB levels within tumor tissues relative to surrounding non-tumor regions. In addition, tumor penetration depth, lymph node spread, distant metastasis, and tumor staging were closely correlated with ALDOB expression. Their findings imply that ALDOB might function as a potential molecular indicator for gastric cancer when combined with survival analysis results (He et al., 2016). ALDOB may serve a controversial function in various cancer types, according to previous studies. Nevertheless, it is still unclear how ALDOB serves a function in ccRCC.

The main focus of this investigation was to ascertain the potential link between ALDOB level, clinical data, and overall survival (OS) in individuals with ccRCC employing diverse databases, which include The Cancer Genome Atlas (TCGA), Gene Expression Omnibus (GEO), the Human Protein Atlas (HPA) and The University of Alabama at Birmingham Cancer data analysis portal (UALCAN). Also, we examined how ALDOB is correlated with immune-related cells using the Tumor Immune Estimation Resource (TIMER). Finally, we directly assessed the ALDOB protein expression in pathological tissues from ccRCC patients using immunohistochemical staining.

Methods and Materials

Data source

The Cancer Genome Atlas (TCGA), a publicly accessible database containing comprehensive data from a large-scale cancer genome project, provides researchers and academicians with clinicopathological data on 33 distinct cancer types. By using the TCGA browser, we compiled RNA-Seq data and matched clinicopathological information from 271 individuals with ccRCC exhibiting elevated ALDOB level, and 270 patients with low levels of ALDOB expression. Additionally, we evaluated their correlation based on the clinicopathological characteristics of the individuals. This study was approved by the Ethics Committees of The Fifth Affiliated Hospital, Southern Medical University, with the IRB approval number: 2023-MNWK-K-003.

External validation from the GEO and HPA databases

To verify the differential expression of ALDOB mRNA between ccRCC tissues and adjacent normal tissues, we downloaded the expression profiling data of GSE66271 and GSE53757 from the GEO datasets and extracted the expression data of ALDOB, which were grouped into tumor and normal groups. Then, independent-sample t-test was performed for data analysis, and R package ggplot2 [3.4.4] was used for data visualization. Additionally, protein immunohistochemistry (IHC) data are available for tumors and normal tissues through the Human Protein Atlas (HPA). Our investigation focused on analyzing the ALDOB protein levels in both human cancerous and normal tissues.

Clinicopathological analysis of ALDOB

To address how ALDOB expression was found to be associated with clinical parameters in ccRCC, we obtained and organized RNAseq data and clinical data of the TCGA-KIRC project from the TCGA database, specifically extracting TPM format data. After removing normal samples and those lacking clinical information, the data were processed using log2(value+1). The Kruskal-Wallis test, supported by R packages stats[4.2.1] and car[3.1-0], was then applied to assess the associations between ALDOB mRNA expression and clinical-pathological parameters, including pathological tumor (T) stage, lymph node (N) stage, metastasis (M) stage, and histological grading in ccRCC cases. Data visualization was performed using the ggplot2 [3.4.4] package.

Development and verification of nomograms

The prognostic features encompassing OS, progression-free interval (PFI), and DSS were evaluated utilizing Cox regression and Kaplan–Meier (KM) methodologies within the clinical significance module of the Xiantao platform (https://www.xiantaozi.com). Our computation of the truncation value for the high and low ALDOB expressions was grounded in the median number. The association between clinicopathological characteristics and ALDOB was determined via the Wilcoxon signed rank sum test and logistic regression analysis. Through the utilization of multivariate Cox analyses, we examined survival rates and various clinical characteristics. A P < 0.05 was established as the significance criterion. The multivariate analysis yielded independent prognostic markers, which were subsequently employed to predict the chances of survival over 1, 3, and 5 years. The 45° line represents the best prediction based on comparing predicted probabilities to observed events.

Functional enrichment analysis

The functional enrichment of ALDOB-associated genes was conducted using the clusterProfiler package in R (v3.6.3), including Gene Ontology (GO) terms: biological process (BP), cellular component (CC), and molecular function (MF). GO analysis was performed with a minimum gene count >3, enrichment factor >1.5, and P < 0.01. In addition, Gene Set Enrichment Analysis (GSEA) was performed to identify pathways significantly associated with ALDOB expression. Genes were ranked based on their log2 fold change between high and low ALDOB expression groups. Statistical significance in GSEA was assessed using a permutation test with 1,000 permutations, which is a commonly used default. In this procedure, sample or gene labels are randomly shuffled 1,000 times to generate a null distribution of enrichment scores (ES), against which the observed ES is compared to calculate the P-value. More permutations lead to more accurate P-value estimation but also increase computation time. Pathways were considered significantly enriched with an adjusted P < 0.05 and false discovery rate (FDR) <0.25. Normalized enrichment scores (NES), adjusted P-values, and FDR were used to report enrichment significance. GSEA enrichment analysis and visualization were conducted utilizing the Cluster Profiler package (Yu et al., 2012).

Immune infiltration analysis

Bindea G’s study (Bindea et al., 2013) identified marker genes for 24 different kinds of immune cells, which were subsequently analyzed for infiltration into tumors using ssGSEA. Using Spearman correlation analysis, we studied the immune cell infiltration in cohorts exhibiting elevated and reduced ALDOB levels and examined the association of ALDOB with the 24 immune cell types. We applied the immune infiltration module of the “Xiantao tool” (https://www.xiantaozi.com; accessed on 1 January 2023) to complete the above analysis.

TIMER database analysis

Tumor Immune Estimation Resource (TIMER2.0) database was utilized for further analysis of immune infiltration in different cancer types, offering a user-friendly online platform for such investigations (https://timer.cistrome.org/) (Li et al., 2020). A total of 10,897 specimens from the TCGA database comprising 32 types of cancer were evaluated for immune internal infiltrates. Moreover, the database is capable of accurately estimating tumor purity as well.

Tissue microarray and immunohistochemistry

ALDOB expression was measured using microarrays comprising 80 ccRCC tissues and 80 adjacent normal tissues (Zhuoli Biotechnology Co, Shanghai, China). In summary, after using a normal serum to block the slides, they were incubated with an anti-ALDOB antibody from Proteintech and left at 4 °C for the night. After that, peroxidase-conjugated avidin-biotin complex and biotinylated secondary antibodies were added (Zhuoli Biotechnology Co, Jiaozhu, China). 3,3-diaminobenzidine (Zhuoli Biotechnology Co., Jiaozhu, China) was utilized as a chromogen for visualizing the immunostaining, with hematoxylin serving as a counterstain. The H-Score method was employed to measure the intensity of ALDOB staining, computed as H-Score = ∑ (pi × i) = (% of weak intensity × 1) + (% of moderate intensity × 2) + (% of strong intensity ×3). The collection and use of pathological tissues were obtained with written informed consent from the patients.

Results

Tumor samples have lower levels of ALDOB expression than normal tissues

The TCGA dataset was utilized to determine whether ALDOB’s low expression is widespread in cancer. Cancers such as cholangiocarcinoma (CHOL), esophageal carcinoma (ESCA), and kidney renal papillary cell carcinomas (KIRP) display down-regulated levels of ALDOB expression, whereas colon adenocarcinoma (COAD) and rectum adenocarcinoma (READ) produce higher levels (Figs. 1A–B). TCGA databases were used to predict ALDOB mRNA levels in 539 samples of ccRCC and 72 samples of normal tissue. CcRCC samples expressed markedly lower levels of ALDOB mRNA than normal tissue (Fig. 1C, P < 0.001). It was also found that ALDOB expression is diminished in adjacent ccRCC samples relative to GTEx samples (Fig. 1D, P < 0.001). A notable decrease in ALDOB expression was noted in 72 ccRCC samples relative to adjacent matched samples (Fig. 1E, P < 0.001). A receiver operating characteristic (ROC) curve was executed to estimate the diagnostic value of ALDOB levels. The AUC for ALDOB levels was 0.836 (CI = 0.777−0.895), showing promising diagnostic potential (Fig. 1F). Further, normal tissues showed an elevation in ALDOB protein levels when compared to ccRCC tissues (Fig. 1G).

Figure 1 The expression profle of ALDOB in ccRCC.

(A–B) Comparison of ALDOB expression in various human cancer tissues with that in normal samples; (C) Decreased ALDOB level was observed in KIRC tissues in comparison to normal tissues; (D) Difference in ALDOB levels in KIRC and adjoining normal samples from the GTEx; (E) KIRC tissues showed higher levels of ALDOB expression compared to the corresponding normal tissues (n = 72); (F) Analyzing the ROC curve of ALDOB in patients with KIRC; (G) The ALDOB protein levels were markedly increased in non-paired normal tissues as opposed to KIRC tissues (*p < 0.05, **p < 0.01, ***p < 0.001).

Additionally, the ALDOB gene levels were verified through examination of the GEO datasets (GSE66271 and GSE53757). The results were consistent with the data from TCGA, the ALDOB level was diminished in ccRCC tissues versus adjacent normal tissues (Figs. 2A, 2B). HPA data also showed that ccRCCs exhibited a decrease in ALDOB expression compared to normal tissues (Fig. 2C). Based on the above data, ALDOB appears to be an important diagnostic biomarker in ccRCC tissues.

Figure 2 Evaluating the expression of ALDOB utilizing the Gene Expression Omnibus datasets and the Human Protein Atlas (HPA).

(A) Assessment of reduced ALDOB mRNA levels in clear cell renal cell carcinoma (ccRCC) versus adjacent healthy tissues utilizing the GSE66271 dataset. (B) Confirmation of diminished ALDOB mRNA abundance in ccRCC relative to non-cancerous specimens within the GSE53757 dataset. (C) Analysis of HPA information revealed that ALDOB protein quantities in renal cell carcinoma specimens were diminished versus healthy kidney tissue (utilizing antibody HPA002198, HPA073201, and 10X) (**p < 0.01, and ***p < 0.001).

Clinical characteristics associated with ALDOB expression

We assessed whether ALDOB was correlated with different clinical-pathological features, which include pathologic T stage (T1, T2, T3 and T4), pathologic N stage (N0 and N1), pathologic M stage (M0 and M1), histologic grade (grade 1, 2, 3, and 4), and pathologic stage (stages I, II, III, and IV) by applying TCGA dataset. Statistical evaluation through logistic regression confirmed the link between ALDOB and the clinical-pathological features of individuals with ccRCC utilizing the TCGA-KIRC cohort. The findings demonstrated that individuals with ccRCC who were upgraded in the T stage, N stage, M stage, pathologic stage, and histologic grade exhibited decreasing ALDOB levels. ALDOB was found to be overexpressed in the patients with T1 stage (Fig. 3A), no lymph node metastases (Fig. 3B) or no distant metastases (Fig. 3C), pathologic stage I (Fig. 3D), and histologic grade 1 (Fig. 3E), respectively. As a result of analyzing the expression of ALDOB in male and female ccRCC patients, it was determined that males had a relatively lower expression of ALDOB than females (Fig. 3F).

Figure 3 Clinical and pathological parameters of the KIRC correlated with ALDOB mRNA expression levels.

(A) Pathologic T stage; (B) Pathologic N stage; (C) Pathologic M stage; (D) Pathologic stage; (E) Histologic grade; (F) Gender. G1, grade 1; G2, grade 2; G3, grade 3; G4, grade 4 (*p < 0.05, **p < 0.01, and ***p < 0.001).

In addition, the Fisher exact test and the chi-square test yielded consistent results (Table 1). Subsequently, several clinical variables were markedly correlated with ALDOB expression according to the univariate logistic regression analysis, which include pathologic stage (odds ratio (OR) = 0.662 [0.464−0.658], P = 0.023), histologic grade (OR = 0.629 [0.447−0.886], P = 0.008), gender (OR = 0.645 (0.452−0.922), P = 0.016), and pathologic stage (OR = 0.668 [0.471−0.948], P = 0.024) (Table 2). Nonetheless, no marked variations were observed in pathologic N stage (OR = 0.368 (0.115−1.174), P = 0.091), and pathologic M stage [OR = 0.623 (0.383−1.015), P = 0.057] (Table 2). According to these findings, ALDOB might contribute to tumor development and progression in individuals with ccRCC.

Table 1 Relationship between ALDOB expression and clinicopathological characteristics in patients with KIRC.

Characteristics	Low expression of ALDOB	High expression of ALDOB	P value	
n	270	271		
Pathologic T stage, n (%)			<0.001	
T1	118 (21.8%)	161 (29.8%)		
T2	44 (8.1%)	27 (5%)		
T3	99 (18.3%)	81 (15%)		
T4	9 (1.7%)	2 (0.4%)		
Pathologic N stage, n (%)			0.080	
N0	127 (49.2%)	115 (44.6%)		
N1	12 (4.7%)	4 (1.6%)		
Pathologic M stage, n (%)			0.056	
M0	205 (40.4%)	224 (44.1%)		
M1	47 (9.3%)	32 (6.3%)		
Gender, n (%)			0.016	
Female	80 (14.8%)	107 (19.8%)		
Male	190 (35.1%)	164 (30.3%)		
Age, n (%)			0.699	
≤60	132 (24.4%)	137 (25.3%)		
>60	138 (25.5%)	134 (24.8%)		
Histologic grade, n (%)			<0.001	
G1	4 (0.8%)	10 (1.9%)		
G2	105 (19.7%)	131 (24.6%)		
G3	102 (19.1%)	105 (19.7%)		
G4	54 (10.1%)	22 (4.1%)		
Serum calcium, n (%)			0.013	
Low	90 (24.5%)	114 (31.1%)		
Normal	91 (24.8%)	62 (16.9%)		
Elevated	4 (1.1%)	6 (1.6%)		
Pathologic stage, n (%)			0.002	
Stage I	115 (21.4%)	158 (29.4%)		
Stage II	37 (6.9%)	22 (4.1%)		
Stage III	65 (12.1%)	58 (10.8%)		
Stage IV	50 (9.3%)	33 (6.1%)		

Table 2 Logistic regression analysis of ALDOB expression.

Characteristics	Total (N)	OR (95% CI)	P value	
Pathologic T stage (T3&T4 vs. T1&T2)	541	0.662 (0.464–0.944)	0.023	
Pathologic N stage (N1 vs. N0)	258	0.368 (0.115–1.174)	0.091	
Pathologic M stage (M1 vs. M0)	508	0.623 (0.383–1.015)	0.057	
Gender (Male vs. Female)	541	0.645 (0.452–0.922)	0.016	
Histologic grade (G3&G4 vs. G1&G2)	533	0.629 (0.447–0.886)	0.008	
Pathologic stage (Stage III&Stage IV vs. Stage I&Stage II)	538	0.668 (0.471–0.948)	0.024	

Prognostic value of ALDOB expression in ccRCC

Subsequently, we generated an OS heatmap for the ALDOB gene. The analysis revealed a significant association between low ALDOB expression and overall survival (OS) in multiple tumor types, including COAD, KIRC, and LGG (Fig. 4A). Data from the TCGA database was utilized to examine the link between ALDOB level and prognostic outcomes (OS, DSS, PFI). Low ALDOB expression was linked to unfavorable OS, DSS, and PFI in the study population, with hazard ratios (HR) of 0.49 (95% CI [0.36–0.66]), 0.37 (95% CI [0.24–0.55]), and 0.47 (95% CI [0.34–0.66]), respectively, all with a p-value <0.001 (Figs. 4B–4D). A further investigation of ALDOB expression and subgroups was conducted in this study. Lower expression of ALDOB was detected in the T3-T4 stage (HR = 0.49 (0.33–0.73), P < 0.001), pathologic stage III–IV (HR = 0.51 (0.35–0.74), P < 0.001), histologic grade G3-G4 (HR = 0.48 (0.33–0.68), P < 0.001) (Fig. 4F). We developed a clinical prognostic risk score using the stages T, N, M, pathologic stage, histologic grade, and ALDOB expression in ccRCC (Fig. 4E). Utilizing Cox logistic regression analyses outcomes, these prognostic indicators were integrated to construct nomograms predicting 1-year, 3-year, and 5-year OS for ccRCC patients within TCGA-KIRC dataset (Fig. 4G). Furthermore, a calibration chart was employed to evaluate the model’s predictive accuracy (Fig. 4H). According to the findings, ALDOB expression levels could better predict patient survival at 3- and 5-year intervals.

Figure 4 Expression of ALDOB as a prognostic indicator.

(A) Establishment of an OS heatmap of the ALDOB gene. (B–D) Results indicated a significant ALDOB downregulation in patients with poor prognoses relative to those with high expression levels, as evidenced by OS, DSS, and PFI outcomes (P < 0.001). (E) Risk scores and survival status of ALDOB gene in KIRC patients. (F) Evaluation of the prognosis associated with ALDOB expression in clinical subgroups. (G) A nomogram was developed utilizing the clinical characteristics of ALDOB expression. (H) Multivariate Cox regression calibration chart displays the model’s predictive ability.

Subsequently, the univariate and multivariate Cox regression analyses were executed to further ascertain the independent predictors of OS, DSS, and progression-free survival (PFS) in individuals with ccRCC. The forest plots indicated that ALDOB level exhibited marked links to OS (HR = 2.065, 95% CI [1.330–3.205], P = 0.001) (Fig. 5A), DSS (HR = 2.352, 95% CI [1.340–4.129], P = 0.003) (Fig. 5B), and PFS (HR =2.003, 95% CI [1.268–3.163], P = 0.003) (Fig. 5C) in ccRCC. Further analysis of OS outcomes among patients with different phenotypes between low and high ALDOB expression subgroups was conducted. Moreover, the analysis illustrated that ALDOB downregulation was correlated with worse OS outcomes in patients with T3 stage (p = 0.001), histologic grade 3 (p = 0.028), N0 stage (p = 0.001), M0 stage (p = 0.006), and M1 stage (p = 0.010) (Fig. 6). According to these findings, ALDOB expression levels are associated with prognosis in ccRCC.

Figure 5 Forest plots of univariate and multivariate Cox regression analysis of factors affecting the survival of ccRCC patients from the TCGA-KIRC dataset.

(A) OS; (B) DSS; (C) PFI.

Figure 6 Survival curves for KIRC patients stratified by various clinical characteristics among those with high and low ALDOB expression.

Kaplan–Meier survival for OS among groups stratified by (A) pathologic T3 stage, (B) pathologic N0 stage, (C) pathologic M0 stage, (D) pathologic M1 stage, (E) pathologic stage III/IV, (F) histologic grade 3, (G) low hemoglobin and (H) low serum calcium in KIRC patients.

GO/GSEA enrichment analysis related to ALDOB gene expression in ccRCC tissue

A gene expression profiling analysis was undertaken to understand ALDOB’s biological significance in ccRCC. The analysis identified 2,256 downregulated and 482 upregulated genes that were substantially connected with ALDOB level (Padj < 0.05 and logFC > 1) (Fig. 7A). Additionally, GO and KEGG pathway examinations of ALDOB and its associated genes were executed to explore ALDOB’s functional implications in ccRCC. The findings indicated that within ccRCC, ALDOB primarily functioned in the basal part of the cell (cellular component), serine-type peptidase activity (molecular function), and neuroactive ligand–receptor interaction (KEGG pathway) (Table 3).

Figure 7 GO/GSEA enrichment analysis related to ALDOB gene expression in ccRCC.

(A) GO enrichment analysis was conducted on differentially expressed genes identified through ALDOB expression screening. (B–E) Gene set enrichment analysis results for ALDOB.

We performed GSEA to find functional and biological pathways between low and high ALDOB levels. According to the normalized enrichment scores (NESs), the pathway with the highest enriched signaling was selected based on the expression of the ALDOB gene (Fig. 7B). GSEA indicated that the phenotype associated with low ALDOB expression was primarily enriched in KEGG_PPAR_SIGNALING_PATHWAY (Fig. 7C, NES = 2.599, P < 0.001). WikiPathways_PPAR_SIGNALING_PATHWAY (Fig. 7D, NES = 2.599, P < 0.001). And KEGG_FATTY_ACID_METABOLISM (Fig. 7E, NES = 2.625, P < 0.001).

Association of immune infiltration with ALDOB expression

Subsequently, 24 different immune cells were evaluated for correlation with ALDOB levels in ccRCC. The expression of ALDOB demonstrated significant correlations with neutrophils (r = 0.295), and Th17 cell (r = 0.238), while exhibiting adverse associations with Treg (r =  − 0.160), and macrophages (r =  − 0.133), with all p-values < 0.001 (Figs. 8A–8B). Subsequent analysis revealed notable variations in ALDOB expression levels across various infiltrating immune cell types, such as neutrophils, TReg, and TH17 cells (Figs. 8C–8E). These findings suggest a pivotal role for ALDOB in immune infiltration within ccRCC.

Expression level of ALDOB in tissue microarray samples

We further obtained 80 pairs of ccRCC clinical tissues from tissue microarray (cancerous tissues and paracancerous tissues). IHC showed that the ALDOB level in ccRCC samples was notably decreased relative to that in the adjacent normal tissues, which was confirmed using the corresponding IHC Score (p < 0.05, Figs. 9A–9B). Additionally, we also examined the difference of ALDOB expression in tissues of renal clear cell carcinoma with different pathological stages, and found that the ALDOB expression in tumor tissues of stage II/III was markedly lower than that of stage I/II (p < 0.001, Fig. 9C). ALDOB expression in stage T2 tumors was markedly lower than that in stage T1 tumors (p < 0.05, Fig. 9D). It was found that the expression of ALDOB in AJCC stage II tumors was markedly diminished relative to the AJCC stage I (p < 0.05, Fig. 9E).

Discussion

ccRCC, a type of renal cell carcinoma with the highest prevalence in the world, is currently treated with immunotherapy as an innovative approach. Using the TCGA dataset, ALDOB mRNA expression levels in pan-cancer samples were initially analyzed. Several cancers, including ccRCC, have markedly lower levels of ALDOB mRNA than normal tissues. Also, a significant link was noted between the levels of ALDOB mRNA in ccRCC and pathological stage, M stage, histological grade, and N stage. KM survival analysis illustrated that lower ALDOB expression was related to shorter OS, PFI, and DSS, indicating that individuals with lower ALDOB expression may have an unfavorable outcome. It appears that ALDOB is involved in tumor metastasis and progression, making it a potential diagnostic indicator of ccRCC.

The aldolase enzyme family represents the fourth enzymatic step in glycolysis, encompassing ALDOA, ALDOB, and ALDOC, which are products of distinct genetic loci (Penhoet, Rajkumar & Rutter, 1966). Fructose bisphosphate aldolase B, alternatively termed aldolase B, originates from the ALDOB gene and demonstrates predominant expression within hepatic and renal tissues (Tang & Cui, 2024). Ongoing investigations have examined ALDOB’s prognostic significance across diverse cancer types and explored its mechanistic contributions to tumor development. Research by Xia et al. (2021) demonstrated that ALDOB functions as an autonomous predictor of metastasis-free survival in prostate cancer patients, exhibiting significant correlations with both pathological grading and tumor staging. Multiple investigations have established ALDOB’s regulatory role in cancer progression through its involvement in crucial signaling cascades, including the PI3K/AKT/mTOR axis, GSK-3β pathway, and Wnt signaling network (He et al., 2020; Liu, Hu & Jin, 2025; Liu et al., 2024). Previous studies have found that in obese mice, ALDOB acts as a core protein linking the cluster of proteins involved in xenobiotic metabolism with those involved in fatty acid metabolism and the PPAR signaling pathway (Nesteruk et al., 2014). Moreover, it may influence PPAR-mediated fatty acid oxidation, mitochondrial function, and energy metabolism. Consistently, our GSEA analysis also revealed that ALDOB regulates the progression of ccRCC through PPAR signaling transduction.

Table 3 GO enrichment analysis results.

Ontology	ID	Description	GeneRatio	BgRatio	p value	p adjust	
CC	GO:0016323	basolateral plasma membrane	12/191	226/19594	2.33e–06	0.0006	
CC	GO:0009925	basal plasma membrane	12/191	251/19594	6.86e–06	0.0009	
CC	GO:0045178	basal part of cell	12/191	269/19594	1.38e–05	0.0012	
CC	GO:0034364	high-density lipoprotein particle	4/191	27/19594	0.0001	0.0082	
CC	GO:0016324	apical plasma membrane	12/191	358/19594	0.0002	0.0109	
MF	GO:0015301	anion:anion antiporter activity	5/178	24/18410	2.92e–06	0.0005	
MF	GO:0140323	solute:anion antiporter activity	5/178	24/18410	2.92e–06	0.0005	
MF	GO:0004252	serine-type endopeptidase activity	9/178	174/18410	5.02e–05	0.0061	
MF	GO:0008236	serine-type peptidase activity	9/178	191/18410	0.0001	0.0089	
MF	GO:0017171	serine hydrolase activity	9/178	195/18410	0.0001	0.0089	
KEGG	hsa04721	Synaptic vesicle cycle	6/72	78/8164	5.93e–05	0.0042	
KEGG	hsa04080	Neuroactive ligand–receptor interaction	12/72	362/8164	6.65e–05	0.0042	
KEGG	hsa04966	Collecting duct acid secretion	4/72	27/8164	8.37e–05	0.0042	
KEGG	hsa05323	Rheumatoid arthritis	5/72	93/8164	0.0013	0.0492	

Figure 8 Analysis of the correlation between immune infiltration and ALDOB expression.

(A) The association of the expression of ALDOB with 24 immune cells. (B) Superimposed bar graph of immune infiltration of 22 immune cells in the high-expression and low-expression subgroups of ALDOB. (C–E) Infiltration of specific immune cells is correlated with ALDOB expression. (ns) indicates P ≥ 0.05; *P < 0.05; **P < 0.01; ***P < 0.001; ****P < 0.0001.

Figure 9 Protein expression level of ALDOB in KIRC tissues and their matched normal tissues.

(A) Illustrations of normal and KIRC tissues stained with immunohistochemistry. (B) The IHC scores of ALDOB in KIRC and normal tissues. (C) The IHC scores of ALDOB in pathologic stage I/II and II/III tissues (The pathologic stage is determined according to the 2016 WHO/ISUP grading classification). (D) The IHC scores of ALDOB in pathologic T1 stage and T2 stage tissues. (E) The IHC scores of ALDOB in AJCC stage II and AJCC stage I tissues.

The findings from this investigation demonstrated that ALDOB functioned as an autonomous predictive indicator in determining ccRCC patient outcomes. Subsequently, predictive models with enhanced prognostic capabilities were developed to estimate OS, DSS, and PFS in individuals with ccRCC by combining multiple parameters: T stage, N stage, M stage, histologic grade, and ALDOB levels. These predictive frameworks enable healthcare providers to evaluate ccRCC clinical outcomes with greater precision and establish individualized prognostic evaluation approaches for patients. Multivariate Cox regression examination revealed ALDOB as a standalone prognostic indicator for OS, DSS, and PFS among ccRCC patients. Notably, individuals exhibiting elevated ALDOB expression demonstrated more favorable outcomes, which can be partially explained by their correlation with reduced T stage, N stage, M stage, and histologic grade classifications.

This study also found that ALDOB strongly relates to the level of infiltrating immune cells in ccRCC. As a crucial component of cancer development, infiltration of immune cells in the tumor microenvironment (TME) may affect the efficiency of chemotherapy, immunotherapy, and radiotherapy, thus affecting cancer patient outcomes (Wang et al., 2024; Yin et al., 2025). Nonetheless, it is uncertain whether immune infiltration in ccRCC is correlated with ALDOB expression. To assess whether ALDOB expression correlates with ccRCC immune infiltration, we conducted a systematic analysis. For the first time, we identified ALDOB as a regulator of immune infiltration in ccRCC. Additionally, our findings suggested that decreased levels of ALDOB were not only linked to neutrophils and Th17 cells but also had a strong link to Treg cells grounded in the results of immune infiltration analysis. As evidenced by these findings, targeting ALDOB could potentially enhance the efficacy of immunotherapy. Overall, ALDOB appears to be implicated in the recruitment and regulation of tumor-infiltrating lymphocytes in ccRCC, warranting further investigation into its molecular mechanisms and impact on the tumor microenvironment.

Recently, a novel concept known as “Immunoscore” has been gaining attention in RCC. Immunoscore is an immune microenvironment-based scoring system designed to evaluate the extent of immune cell infiltration within tumor tissues, thereby predicting patient prognosis and therapeutic response. Currently, Immunoscore has been recognized as an independent prognostic indicator in colorectal cancer, unrelated to TNM staging, where lower Immunoscore values correlate with poorer patient outcomes (Guo et al., 2019; Guo et al., 2020). In RCC, recent studies suggest that integrating Immunoscore with TNM staging and the WHO/ISUP 2016 grading system improves the prediction of disease-free survival (DFS), PFS, and OS in ccRCC patients, highlighting its potential as a valuable prognostic tool following nephrectomy (Selvi et al., 2020). Another study emphasized that while Immunoscore also predicts OS in non-ccRCC, its prognostic performance is slightly lower compared to ccRCC, indicating significant immunological differences between ccRCC and non-ccRCC subtypes (Selvi et al., 2021). This finding underscores the importance of further investigating these immunological disparities to optimize Immunoscore application across different RCC subtypes.

Moreover, our supplementary investigations revealed that ALDOB functions as a crucial prognostic indicator across multiple cancer varieties (Huang et al., 2022; Li et al., 2017; Tian et al., 2017; Zhao & Xu, 2023). Building on these observations, we employed the TIMER platform to examine ALDOB expression distributions among diverse cancer classifications. Notably, within ccRCC, ALDOB mRNA levels demonstrated a substantial reduction in cancerous tissues when contrasted with healthy kidney specimens. This examination additionally established that ALDOB protein expression was diminished in ccRCC samples relative to surrounding normal tissue, with more pronounced decreases observed in pathologic stage II/III patients, pathologic stage T2 patients, and AJCC stage II patients.

Despite uncovering the possible impact of ALDOB on the immune infiltration and prognosis in ccRCC, our study is still constrained by various limitations. One limitation of this study is that the data primarily comes from online platform databases, which are subject to continuous updates and expansions, potentially impacting the research results. Second, the validation of ALDOB function in ccRCC and its molecular mechanism in ccRCC immunity has not been conducted through in vitro and in vivo studies. To further validate the projected outcomes, experiments will be conducted in a follow-up study.

Conclusion

Overall, our findings highlight that ALDOB is prominently expressed in ccRCC tissues, and its downregulation is strongly linked to worse survival outcomes in ccRCC. Additionally, ALDOB expression is implicated in the regulation of neutrophils, and Treg cells. ALDOB could emphasize its unique possible role in modulating the infiltration of immune cells in individuals with ccRCC.

Supplemental Information

Supplemental Information 1 ALDOB tissue microarrary results

Supplemental Information 2 Dali Wu’s contributions

We thank Bullet Edits Limited for the linguistic editing and proofreading of the manuscript. The Xiantao tool was used for differential gene expression analysis, GO and KEGG enrichment analysis, immune cell infiltration analysis, and survival analysis. The platform provided computational support through a graphical interface, enabling reproducible and systematic analysis without the need for programming.

Additional Information and Declarations

Competing Interests

Author Contributions

Human Ethics

Microarray Data Deposition

Data Availability

The authors declare there are no competing interests.

Wu Xu performed the experiments, authored or reviewed drafts of the article, and approved the final draft.

Dali Wu conceived and designed the experiments, performed the experiments, analyzed the data, prepared figures and/or tables, authored or reviewed drafts of the article, and approved the final draft.

Cuilian Li performed the experiments, authored or reviewed drafts of the article, and approved the final draft.

Lingfei Yan analyzed the data, prepared figures and/or tables, and approved the final draft.

Bo Peng analyzed the data, authored or reviewed drafts of the article, and approved the final draft.

Yang Luo conceived and designed the experiments, analyzed the data, prepared figures and/or tables, and approved the final draft.

Dawei Liu analyzed the data, prepared figures and/or tables, and approved the final draft.

Qing Li conceived and designed the experiments, authored or reviewed drafts of the article, and approved the final draft.

Tao Wang conceived and designed the experiments, prepared figures and/or tables, and approved the final draft.

The following information was supplied relating to ethical approvals (i.e., approving body and any reference numbers):

The Ethics Committees of The Fifth Affiliated Hospital, Southern Medical University, approved the study (2023-MNWK-K-003).

The following information was supplied regarding the deposition of microarray data:

Data is available at NCBI GEO, accession numbers: GSE284424.

The following information was supplied regarding data availability:

Due to the issue of release time, the data was not found. Currently, the data has been published in NCBI GEO, accession numbers: GSE284424.

Code is available at Zenodo:

WANG, T. (2025). R code for analysis. Zenodo. https://doi.org/10.5281/zenodo.15479235.

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
