# Peer review of "ALDOB is a prognostic biomarker and a potential immunotherapy target for clear cell renal cell carcinoma"

_PeerJ, doi:10.7717/peerj.19869_

## Round 0.1 · original submission · Major Revisions

· Academic Editor

Major Revisions

Figure 1 needs to have some of the text made larger/bolder for visibility (as one example, panel G; Same with Figure 4A), and figure legends should be expanded to include information about individual panels, statistical tests, colors used, etc. Finally, gene and protein names should follow standard italics/capitalization convention.

Reviewer 1 ·

Basic reporting

no comment

Experimental design

no comment

Validity of the findings

no comment

Annotated reviews are not available for download in order to protect the identity of reviewers who chose to remain anonymous.

Reviewer 2 ·

Basic reporting

The paper contains several issues with sentence structure, making some sections unclear or difficult to follow. Additionally, there are multiple typos and areas where wording could be improved for clarity and conciseness. I have highlighted specific instances in the attached PDF for your review.

Additionally, some references are missing, such as the lack of citation for the ideas presented in line 240. Ensuring proper citation will strengthen the credibility of the arguments.

Experimental design

The research question being addressed is valid and relevant. Given that different cancers exhibit varying aldolase B levels, it is important to determine its level in ccRCC.

the methods section is not well defined and, in some cases, is mixed with the results, such as in lines 67 and 87. Clearly distinguishing the methods from the results will improve the structure and readability of the paper.

Validity of the findings

It is unclear whether a sample size calculation was performed and how the numbers listed in lines 76 and 122 were determined. Without this information, the reliability and accuracy of the results may be affected.

Annotated reviews are not available for download in order to protect the identity of reviewers who chose to remain anonymous.

---

## Round 0.2 · Minor Revisions

· Academic Editor

Minor Revisions

Thank you for addressing the prior reviews. Please add the publicly available code DOI for analysis replication as required by the journal.

Additionally, some phrasing in the manuscript is still confusing and critical details are still missing to ensure transparency and reproducibility. For example, 'with the genome being ranked 1000 times per analysis' in the section on functional enrichment analysis, detailed methods explaining how differential expression was verified (2.2), how expression was found to be associated with clinical parameters (2.3), typos like a missing space between 45 and line (2.4), versioning information and/or access date information for the xiantao platform, etc.

Reviewer 1 ·

Basic reporting

The paper is clear and unambiguous, with professional English used throughout. Literature references, sufficient background/context provided. Professional article structure, figures, and tables used. Raw data is shared. Hypotheses are self-contained with relevant results.

Experimental design

The research question is well defined, relevant and meaningful. Explains how the research fills an identified gap in knowledge. Rigorous investigation conducted to a high technical & ethical standard. Methods described with sufficient detail & information to allow replication.

Validity of the findings

Impact and novelty not assessed. Meaningful replication is encouraged where the rationale & contribution to the literature is clearly stated. All underlying data are provided; they are robust, statistically sound & controlled. Conclusions are well stated, linked to the original research question & limited to supporting findings.

Additional comments

I would like to congratulate the authors on the development of the article after the revisions they made according to the reviewer's suggestions. I believe that the article will contribute to the literature in this form.

---

## Round 0.3 · Minor Revisions

· Academic Editor

Minor Revisions

With respect to code, as one example, section 2.3 references R packages, so the code supporting these analyses needs to be submitted to a public repository per the journal's policy. Will you please clarify?

---

## Round 0.4 · accepted · Accept

· Academic Editor

Accept

Thank you for addressing the comments. This manuscript is now ready for publication.